# Treatment of Lichen Planopilaris and Frontal Fibrosing Alopecia: A Retrospective, Real-Life Analysis in a Tertiary Center in Germany

**DOI:** 10.3390/jcm13164947

**Published:** 2024-08-22

**Authors:** Henner Stege, Maximilian Haist, Michael Schultheis, Johannes Pawlowski, Miriam Wittmann, Stephan Grabbe, Florian Butsch

**Affiliations:** Department of Dermatology, University Medical Center Mainz, 55131 Mainz, Germany

**Keywords:** lichen planopilaris, frontal fibrosing alopecia, systemic treatment, hydroxychloroquine

## Abstract

**Background:** Lichen planopilaris (LPP) is an inflammatory cicatricial alopecia characterized by an irreversible destruction of the hair follicle resulting in its permeant destruction. The clinical presentation of LPP is a progressive patchy scarring alopecia. A variety of systemic agents is used to treat LPP with varying success. The aim of this retrospective, real-life analysis was to evaluate the treatment of hydroxychloroquine for LPP. **Method:** In this retrospective, single-center study, we analyzed 110 patients with LPP and frontal fibrosing alopecia (FFA) who received treatment over a 12-month period from March 2014 to March 2021 at the Department of Dermatology, University of Mainz Medical Center. Patient records were analyzed for response to treatment, co-morbidities, disease progression-free survival (DPFS), and safety. Clinical parameters associated with treatment response were determined with Cox regression modelling and logistic regression. **Results:** Overall, 77 of 110 patients were treated with a systemic agent. There was a clear association between LPP and the occurrence of Hashimoto thyroiditis. Topical treatment with corticosteroids did not improve clinical symptoms in the majority of patients (15 out of 101). In 71% of patients treated with systemic cyclosporine A and 62% of patients treated with hydroxychloroquine, we observed a significant resolution of the inflammatory process, which correlated with a robust durable clinical response (*p* < 0.001). Toxicity was observed in 17% (*n* = 9) of patients receiving systemic treatment with hydroxychloroquine and correlated with the duration of systemic treatment (*p* < 0.001). Treatment discontinuation was associated with a flare-up of clinical symptoms (29%), which required the re-initiation of second-line therapy in 13 out of 51 patients. Overall, the initiation of second-line treatment, either hydroxychloroquine or Cyclosporine A (CsA), yielded positive results, especially in the patient cohort treated with hydroxychloroquine (overall response rate, ORR = 100%), who showed disease progression during CsA or retinoids. **Conclusions**: Our results from this contemporary cohort of patients with LPP and FFA indicate that hydroxychloroquine and cyclosporine are effective systemic agents in decreasing clinical symptoms. However, our data also show that the discontinuation of treatment is often associated with the exacerbation of clinical symptoms. Response rates to second-line treatment were especially favorable in the patient cohort with hydroxychloroquine.

## 1. Introduction

Lichen planopilaris (LPP) is histologically defined as cicatricial alopecia caused by chronic lymphocytic inflammation around the isthmus of the hair follicle, eventually resulting in the loss of the epidermal stem cell niche and permanent destruction of the follicle [1]. Similar to other primary cicatricial alopecias which are considered autoimmune diseases, LPP presents with histological features of an autoimmune disease with a prominent involvement of CD8+ T-lymphocytes [2] Thus, the main clinical presentation of LPP is progressive patchy scarring alopecia [2]. Further clinical symptoms may include severe itching and burning, follicular hyperkeratosis or plugging, and perifollicular erythema, depending on the overall inflammatory activity. However, the aetiology and pathogenesis of LPP remain incompletely understood [3]. In 2001, the North American Hair Society proposed a histological classification for primary cicatricial alopecias based on the predominant inflammatory cellular infiltrate, separating them into a lymphocytic, neutrophilic, and mixed group [4].

### 1.1. Lichen Planopilaris

LPP is the predominant type of lymphocytic cicatricial alopecias and is seen more frequently in Caucasian women than in other populations. LPP usually presents with multifocal alopecia patches with perifollicular erythema and follicular hyperkeratosis at the hair-bearing margin which are located at the parietal capillitium, as well as the forehead [1]. In addition, in about 20% of LPP cases, cutaneous or mucosal lichen planus lesions may co-exist. Diagnosis is often made in consideration of clinical symptoms and histopathological findings [5]. Histopathology from active LPP lesions often shows a perifollicular lymphocytic inflammation (interface dermatitis) involving the follicular infundibulum and isthmus, as well as perifollicular fibrosis and cytoid bodies [6]. Comorbidities associated with LPP are an excess of androgen and thyroid disease, especially in women [7].

### 1.2. Frontal Fibrosing Alopecia

Frontal fibrosing alopecia (FFA) is considered a variant of LPP and was first described by Kossard in 1994 [8]. The main clinical presentation of FFA is a progressive recession of the frontotemporal hairline due to a scarring hair loss. Alongside this, a partial or complete loss of eyebrows is also a common clinical feature, which may precede, accompany, or follow the hairline recession. Since the FFA is considered a sub-variant of LPP, their histologopathological characteristics are largely indistinguishable. FFA affects almost exclusively women and the first clinical symptoms typically occur during menopause or are postmenopausal [9,10].

The therapeutic goal in LPP is the reduction of inflammatory activity, thus preventing further scarring and subsequent hair loss. Different strategies have been applied to treat LPP depending on the severity and activity of the disease, via topical or systemic agents, or a combination of both [11]. In mild to moderate cases, topical corticosteroids (tCS) and topical calcineurin inhibitors (tCI) are most commonly used as first-line agents. A rapid disease progression or extended scalp involvement may require the initiation of systemic treatment regimens. Accordingly, immunomodulating agents such as hydroxychloroquine can be utilized to treat LPP. The immunomodulatory effects of hydroxychloroquine at the cellular level are a result of the inhibition of lysosomal activity and autophagy and a decrease in the secretion of pro-inflammatory cytokines. However, there are limited data regarding the efficacy of those treatment regimens [12,13,14]. Thus far, the treatment of LPP is mainly empirical and based on small case series and consensus opinions. Given the lack of information from large-scale randomized trials comparing the different treatment regimes in patients with LPP/FFA, it remains unclear which systemic treatment might be best for patients with LPP. In this single-center retrospective analysis, we, therefore, aimed to determine the efficacy of the treatment with hydroxychloroquine, acitretin, or other immunomodulatory agents in reducing clinical symptoms of LPP in our real-world patient cohort and to identify factors that could potentially be used to guide decisions regarding treatment decisions.

## 2. Patients and Methods

### 2.1. Patient Population

In this retrospective, single-center study, we report on the clinical outcomes of patients with LPP or FFA who received either topical and or systemic treatment at the University Medical Center Mainz between March 2014 and March 2021 with follow-up until December 2021. Patients were eligible for analysis if the diagnosis was based on the clinical presentation and/or biopsy reports with a clinical correlation of LPP or FFA and if they received topical or systemic treatment for at least three months and a follow-up of 6 months was possible. We identified 110 patients (18 male and 92 female) who received either topical or systemic treatment. Data cut-off was set for December 2021 (see Figure 1).

Data on baseline demographics, duration, and extent of the disease, clinical findings (symptoms and signs of the disease), front-line and second-line treatments (i.e., treatment regimen, treatment duration, treatment cessation due to adverse events (AEs), and disease progression), and laboratory studies were collected by electronic chart review.

### 2.2. Clinical Outcomes

We analyzed the impact of different treatment regimens on clinical outcomes of patients. Therefore, we stratified patients into two cohorts: those who had a topical treatment before receiving a systemic agent or those who received as a first-line treatment a systemic agent in combination with a topical treatment. The primary clinical outcome parameter was defined as response to treatment with a reduction of clinical symptoms of >50% (e.g., reduction of hyperkeratosis, plugging, or itching). Thus, patients with less than 50% reduction in clinical symptoms were considered non-responders. Secondary clinical outcomes included the best overall response (BOR) to second-line therapy and disease progression-free survival (DPFS).

### 2.3. Statistical Analysis

Descriptive statistics were used to analyze the baseline characteristics of the study population. Treatment duration was calculated as the period between initial drug administration and treatment discontinuation. DPFS was calculated from the start of first-line treatment to the date of clinical disease progression or last follow-up. Chi-square test was used to assess the association between the different treatment sequences and clinicopathological features. Confidence intervals (CI) of 95% for categorical variables were calculated using the Clopper–Pearson method. Comparisons between continuous variables of the different treatment sequences were performed using ANOVA variance analysis. 

Cox’s proportional hazards models were applied to identify the strongest predictors for clinical response by adjusting for baseline characteristics, treatment sequence, and laboratory results. Here, hazards ratios (HRs) were provided with 95% confidence intervals (CI). Multivariate analysis was calculated for the significant (*p* ≤ 0.05). In all cases, two-tailed *p*-values were calculated and considered significant with values *p* < 0.05. SPSS (version 27, IBM, Ehningen, Germany), and GraphPad PRISM (Version 5, San Diego, CA, USA) was used for all analyses.

## 3. Results

### 3.1. Patient Characteristics

Of the 189 patients treated with LPP/FFA from March 2014 until March 2021 at the University Medical Center Mainz, 110 patients (18 male and 92 female) were qualified for inclusion into this retrospective analysis. The clinical characteristics of the 110 included patients are summarized in Table 1. In accordance with previous observations, female patients were significantly more often diagnosed with LPP. The age at initial diagnosis ranged between 23–90 years (median 56 years). 

A classic form of LPP was observed in 85 (77%) patients, whereas FFA was evident in 25 (23%) patients. In the majority of the 110 patients, the primary site of fibrosing alopecia was distributed in the parietal area (63%), followed by frontal (28%) and temporal (11%). Hair loss (100%) was the most common clinical sign at presentation, followed by scaling or atrophy (95%) and pruritus (84%). The median time interval from the onset of the first clinical symptoms to the diagnosis of LPP or FFA was 14 months (range: 2–120 months). 

Detailed data on medical records were available in all 110 cases. High blood pressure was recorded in 39 (35.4%) cases, hyperlipidaemia in 19 (17.2%), adult-type diabetes mellitus in 32 (29.0%), and hypothyroidism in 17 (14.9%) patients. Notably, concomitant autoimmune disease was reported in 28 (26%) of the patients, and, of these, nearly all patients reported Hashimoto’s disease (25, 89%). There were no significant laboratory abnormalities (see Table 1).

### 3.2. Topical Treatment

Among the 110 patients, 101 (92%) initially received topical treatment with high-potency topical corticosteroids (tCS) (*n* = 95, 94%) and/or topical calcineurin inhibitors (tCI) (*n* = 24, 22%). The median duration of topical treatment was 10 months (range 2–96 months), and stable disease as the best overall response (BOR) was observed in 15 of these 101 (14%) patients (see Table 2); none of these patients received a systemic treatment down the road. 

The mean time to remission was 6 months (3–17 months). In all patients with stable disease upon initiation of topical treatment, the therapy was ongoing at the time of data-lock. Thus, the patients continued a proactive approach (application twice a week) since the active skin lesions clinically subsided during initial treatment.

### 3.3. Systemic Treatment

Among the 110 patients, 77 (70%) were treated with a systemic agent (see Table 2). No patients treated with a systemic agent responded sufficiently to topical treatment. Systemic treatment was initiated when clinical symptoms slowly progressed over time. The primary reasons why patients refused systemic agents were fear of side effects (60%) and/or no need for systemic therapy (45%). The following medications were most frequently utilized: hydroxychloroquine (200 mg) (*n* = 68, 88%, 55 women and 13 men), followed by cyclosporine A (2.5 mg/kg) (CsA) (*n* = 7, 9%, 7 women) and retinoids (Acitretin 20 mg) (*n* = 2, 3%, 2 women). In particular, 42 (62%) of the patients with first-line hydroxychloroquine treatment showed an initial response upon receiving hydroxychloroquine; similarly, 71% (*n* = 5) of patients with CsA therapy experienced a response, e.g., a reduction in the clinical symptoms. In most patients, the clinical response resulted most often in reduced hair loss (97%) and reduced pruritus (70%). The median duration of first-line treatment in the entire patient cohort was 12 months (range 1–60 months), with 19 (25%) patients still receiving the initial therapy at the time of data cut-off. In detail, the treatment duration for hydroxychloroquine was 17.3 months; for retinoids, 8.5 months; and for CsA, 17.9 months. However, there was no association between the BOR and time from diagnosis to treatment (*p* = 0.70). In response to systemic treatment, 30 patients underwent a tapering out of the systemic treatment with the continuation of topical therapy as a proactive therapy (twice weekly). All patients’ cessation of systemic treatment was due to their wishes.

### 3.4. Response to Hydroxychloroquine and Symptom Relief Is Associated with Treatment Duration

Furthermore, our results revealed that the reduction in hair loss and pruritus correlated with BOR (*p* = 0.037) and the likelihood of a lasting clinical response (*p* < 0.001). As expected, combination treatment (topical plus systemic treatment) improved the overall response (*p* = 0.04). Most patients received corticosteroids and calcineurin inhibitors over the majority of their systemic therapy. Thus, a further analysis of the beneficial impact of either topical corticosteroids or calcineurin inhibitors was feasible.

### 3.5. Renewed Exacerbation of Clinical Symptoms after Discontinuation of System Treatment Is Associated with Time on Systemic Treatment

Overall, 51/68 patients treated with hydroxychloroquine eventually discontinued their first-line treatment. Among these patients, the most common cause for treatment cessation was continued stable disease, due to the patients wishes (26, 51%), toxicity (15, 29%), and the loss of treatment efficacy (10, 18%). Unfortunately, a molecular analysis of the lack of response to therapy was not feasible. When comparing the duration of treatment, we observed a strong correlation between the time on systemic treatment and the likelihood of a renewed exacerbation of clinical symptoms once the treatment was discontinued. Consistent with this, we found that patients who had received systemic treatment with any agent for over 20 months were less likely to relapse after stopping treatment (*p* = 0.03). Different clinical and pathological factors are associated with LPP. When investigating whether these established prognostic factors might impact the response towards systemic treatment (regardless of therapy) using binary regression analysis, we did not observe any significant independent variables. 

### 3.6. Discontinuation of Treatment Was More Likely in Patients Who Experienced Adverse Events

Unfortunately, 17% (*n* = 9) of patients reported treatment-associated adverse events (TAEs), which resulted in the discontinuation of treatment (*p* = 0.008). In addition, we found that TAEs occurred most frequently within the first four months of treatment, resulting in a shorter treatment duration (*p *< 0.001*)* (see Figure 2). TAEs were not associated with gender, or pre-existing autoimmune disorders. Furthermore, TAEs were less likely in patients showing an initial response upon receiving first-line hydroxychloroquine.

### 3.7. Duration of Response and Second-Line Treatment

Upon exacerbation of clinical symptoms (e.g., hair loss and pruritus) after the discontinuation of first-line systemic treatment, 13 of 51 patients received a systemic second-line therapy (see Table 3), while continuing a topical treatment with tCS and/or tCI. Patients with second-line treatment received hydroxychloroquine 32%, methotrexate 15%, CsA 46%, or systemic glucocorticoids 7% (see Figure 3). Notably, the initial response towards first-line treatment was not associated with a response to second-line treatment (*p* = 0.497). Overall, the initiation of a second-line treatment yielded positive results, especially in the patient cohort treated with hydroxychloroquine (overall response rate, ORR = 100%), followed by CsA (66%) and MTX (50%). Interestingly, no patient received systemic second-line treatment with acitretin. Overall, TAEs did not lead to the discontinuation of second-line treatment in any patient. Furthermore, the occurrence of TAEs during first-line treatment did not indicate an increase in TAEs during second-line treatment (*p* = 0.434). In 55% of responders, the stabilization of clinical symptoms led to treatment discontinuation. At the time of data-lock, no disease progression was observed in the patients who discontinued systemic treatment.

## 4. Discussion

The primary aim in treating Lichen planopilaris (LPP) is to reduce chronic inflammation, and, thus, prevent the development of additional alopecic areas [15]. Because LPP is a rare disease, systemic treatment algorithms and comprehensive evidence for either systemic treatment are lacking. Thus, the therapeutic approach towards LPP often results from individual experience, single case reports, and the preference of the treating physician. Furthermore, there are no reliable clinical factors or markers that might predict the response to treatment or rate of remission, or inform on the duration of treatment necessary to achieve remission/minimal disease activity [16,17]. 

Here, we present the outcome of 110 LPP patients treated either with topical or systemic agents. In this retrospective, real-world cohort study, we obtained several findings that contribute to the previous evidence in the topical and systemic treatment of LPP: 

First, baseline patient characteristics and their clinical presentation of LPP were largely comparable with previously published case series or retrospective studies. Accordingly, this study included predominantly females diagnosed with LPP, which showed a wide range of clinical symptoms, from hair loss and pruritus to scaling or atrophy [18,19,20]. Moreover, the loss of eyebrows was more often being observed in patients with FFA as compared to those with classic LPP [13,21]. Most of our patients were initially treated with tCS and topical calcineurin inhibitors. Notably, our data revealed that topical treatment alone did not reduce clinical symptoms, with only 14% of the patients showing lasting therapeutic effects. These data are consistent with the previously published literature, reporting on variable degrees of success for the topical treatment of LPP [22,23]. Overall, our retrospective study indicates that topical treatment alone seldom leads to lasting therapeutic effects. 

Second and most importantly, we demonstrated that hydroxychloroquine, as well as CsA treatment, although in limited usage, in patients with LPP was significantly associated with beneficial treatment outcomes. In our patient population, CsA was only used to a limited extent in the treatment of LPP. Nevertheless, our retrospective data show that a good and robust treatment response can be achieved. With regard to the reduction of clinical symptoms, there was no significant difference between hydroxychloroquine and CsA. In accordance with previous reports, hydroxychloroquine and CsA were equally effective in decreasing clinical symptoms for both LPP and FFA [24]. Additionally, we observed a strong correlation between the duration of systemic treatments and the BOR to first-line treatments. Notably, the duration of treatment was not associated with the rate of TAEs. Moreover, in the case of clinical response, we could not detect significantly more TAEs, but, on the contrary, TAEs were less likely in the response group. Consistent with this finding, our results suggest that patients who ceased first-line therapy were at a significantly higher risk of showing disease progression at any time in the future. 

Another new treatment option is Janus kinase inhibitors (JAKi), a group of small molecules that target the JAK-STAT signaling pathway by inhibiting one or more members of the JAK enzymes in lymphocytes. The rationale for using JAKi in the treatment of LLP is the theory that cytokines activated by the JAK-STAT signaling pathway contribute significantly to the pathogenesis of LP [25]. Therefore, the inhibition of this pathway could potentially lead to positive outcomes, especially in patients with persistent LP [26]. Overall, few data are available on the outcomes of systemic JAKi treatment. In most cases, treatment was initiated as second-line therapy with a treatment duration of almost 11 months, which is consistent with our observations on second-line therapy. In addition, we observed similar response rates to second-line therapy without major toxicities. However, it should be noted that almost all patients with systemic JAKi treatment had previously received hydroxychloroquine or CsA, so JAKi appears to be a viable treatment option when established systemic agents fail to respond in the treatment of LLP or FFA [27,28,29].

The limitations of our study are the retrospective, monocentric nature of the investigation, which adds an inherent selection bias within the cohort. Moreover, the heterogeneity in systemic pretreatments, and subsequent treatment lines, might have affected our results. As itching was not thoroughly documented at the beginning of systemic therapy using the visual analog scale (VAS), we were unable to record the reduction in itching during the systemic treatment. Therefore, a larger, prospective cohort study is needed to validate the observations from our study and confirm the potential value of hydroxychloroquine in the treatment of severe LPP. 

## 5. Conclusions

Our study indicates that a durable response to hydroxychloroquine and CsA can be achieved in most patients with LPP. Furthermore, our observations showed that combination treatment (topical plus systemic treatment) did improve the overall response to systemic treatment and that the likelihood of progress was not associated with combination treatment. Last, our data support a sustained systemic treatment with hydroxychloroquine in the case of no serious TAEs, as treatment termination was strongly associated with disease progression and should, therefore, be avoided. In the case of treatment discontinuation due to TAEs, a re-initiation of hydroxychloroquine or other systemic treatments might present effective treatment modalities.

## Figures and Tables

**Figure 1 jcm-13-04947-f001:**
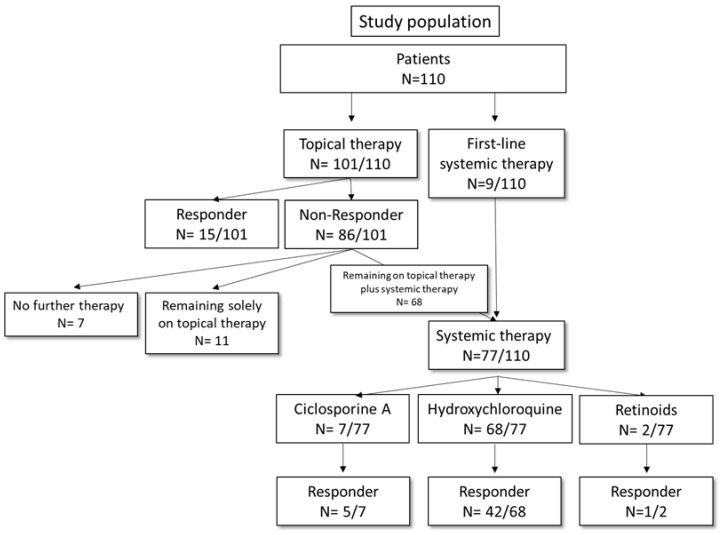
Flow chart illustrating the applied treatment regimens in the LPP patient cohort. Systemic treatments mainly included hydroxycloroquine and, to a small part, CsA or acitretin.

**Figure 2 jcm-13-04947-f002:**
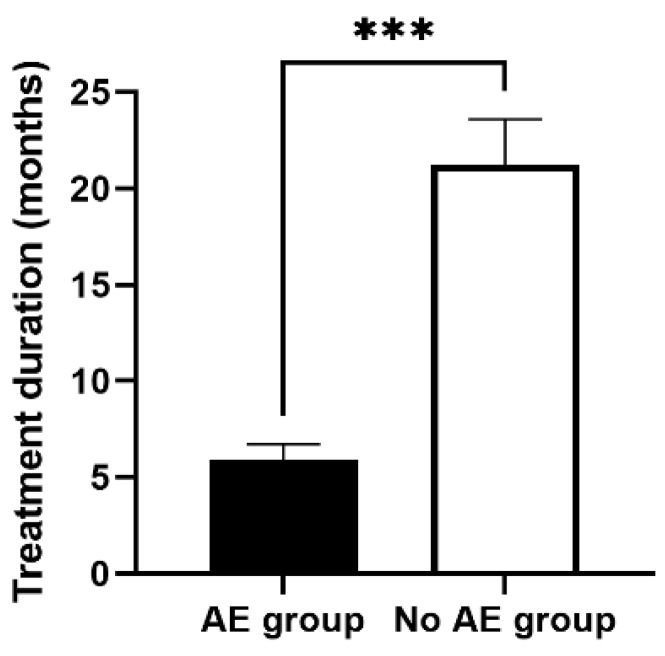
Association of treatment duration and the occurrence of adverse events. Patients who discontinued first-line treatment due to treatment-associated toxicities (TAEs) received therapy for a shorter time period. Abbreviations: *** *p* < 0.001.

**Figure 3 jcm-13-04947-f003:**
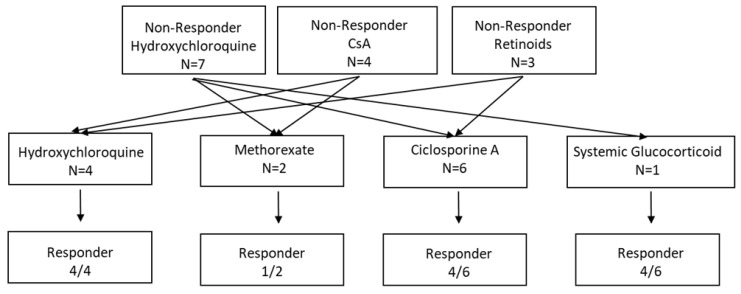
Flow chart illustrating the applied second-line treatment regimens and the response towards the applied treatment in the LPP patient cohort.

**Table 1 jcm-13-04947-t001:** Baseline patient characteristics of the overall patient cohort diagnosed with LPP or FFA.

Baseline Characteristics	
Clinicopathological Features	N (%)
**Total number of patients**	110
Median age at diagnosis	56 (range 23–81)
**Gender**	
Female	92 (84%)
Male	18 (16%)
**Diagnosis**	
Lichen planopilaris (LPP)	85 (77%)
Frontal fibrosing alopecia (FFA)	25 (23%)
**Location of hair loss at presentation**	
Frontal	31 (28%)
Temporal	12 (11%)
Parietal	70 (63%)
Eyebrows	9 (8%)
**Clinical findings on presentation**	
Hair loss	110 (100%)
Perifolicular erythema	42 (38%)
Follicular hyperkeratosis or plugging	39 (35%)
Scaling or atrophy	105 (95%)
Pruritus	92 (84%)
Eyebrow loss	22 (20%)
**Laboratory findings**	
Haemoglobin median	13.7 g/dL (range 11.2–16.9)
Leukocytes median	6.3/nL (range 3.4–13.3)
Thrombocytes median	234/nL (range 33–371)
Low TSH serum levels	11 (10%)
Elevated Aspartate Aminotransferase (AST/GOT) serum levels	9 (8%)
Elevated Alanine Aminotransferase (ALT/GPT) serum levels	8 (7%)
Elevated Gamma-glutamyltransferase (yGT) serum levels	8 (7%)
**Co-morbidities**	
Autoimmune disease	28 (25%)
Hashimoto thyroiditis	24 (86% of patients with AI disease)
Rheumatoid arthritis	4 (14% of patients with AI disease)

**Table 2 jcm-13-04947-t002:** First-line treatments in the overall patient cohort.

First-Line Treatment	N (%)
**Topical**	101 (92% of all patients)
Glucocorticoids III/IV	101 (92%)
Calcineurin inhibitors	24 (22%)
**Outcome**	
Responder	15 (13%)
Non-responder	95 (87%)
**Systemic**	77 (70%)
Hydroxychloroquine 200 mg	68 (88% of systemic Tx)
Retinoide (Acitretin 20 mg)	2 (3%)
Ciclosporine A (5 mg/kg)	7 (9%)
All patients received an additional topical treatment with mild cortisteroid or topical calicneurin inhibitors *	77 (105)
**Outcome**	
Responders	48 (63%)
Hydroxychloroquine	42 (62% of systemic Tx)
Retinoide	1 (50%)
Ciclosporine A	5 (71%)
Non-responders	29 (38%)
Hydroxychloroquine	26 (38%)
Retinoide	1 (50%)
Ciclosporine A	2 (29%)
**Clinical response**	
Reduction of hair loss	47 (97% of responders)
Reduction of perifolicular erythema	14 (29%)
Reduction of follicular hyperkeratosis	36 (75%)
Reduction of pruritus	41 (85%)
**Duration of treatment**	
Median treatment duration in months (range)	12 (range 1–60)
Hydroxychloroquine	17.3
Retinoides	8.5
Ciclosporine	17.9
Ongoing treatment	19 (25%)
Adverse events leading to cessation of treatment	17 (22%)
Acquired secondary resistance	11 (14%)
Cessation of therapy due to stable disease	30 (38%)
Progress after discontinuation	17/58 (29%)

* Due to the regular combination of topical therapy, no further statistical analysis was manageable.

**Table 3 jcm-13-04947-t003:** Second-line treatments and clinical response.

Second-Line Treatment	N (%)
**Second-line therapy**	13
Hydroxychloroquine	4 (32%)
Methotrexate	2 (15%)
Ciclosporine A	6 (46%)
Systemic glucocorticoid	1 (7%)
**Outcome**	
Responders	
Hydroxychloroquine	4 (100%)
Ciclosporine A	4 (66%)
Methotrexate	1 (50%)
Systemic glucocorticoid	1 (100%)
Methotrexate	1 (50%)
Median treatment duration (range)	14 (range 4–96)
Ongoing treatment	4 (31%)
Cessation of therapy due to stable disease	6 (55%)

## Data Availability

All relevant data are within the manuscript.

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
