# Peer review of "Treatment of Lichen Planopilaris and Frontal Fibrosing Alopecia: A Retrospective, Real-Life Analysis in a Tertiary Center in Germany"

_jcm, 2024, doi:10.3390/jcm13164947_

Round 1

Reviewer 1 Report

Comments and Suggestions for Authors

The article "Treatment of Lichen planopilaris and Frontal fibrosing alopecia: A Retrospective, Real-Life Analysis in a Tertiary Center in Germany" may be of interest to both practicing physicians and researchers. It can help to navigate the effective treatment tactics for patients with Lichen planopilaris and Frontal fibrosing alopecia.

I have a few comments on the article:

1. In paragraph 2.1, the authors presented a Figure 1. Flow chart illustrating the applied treatment regimens in the LPP patient cohort.  In the text of the article, the authors describe the results of systemic second-line therapy in patients. It would have been better if the authors had supplemented Fig. 1 with a scheme of systemic second-line treatment. This would make the article easier to read.

2. In the tables, the research results are presented as a percentage. It is difficult to assess it, as they relate to different groups.

3. It should be clarified what the authors meant:

line 22 - (145)???

line 33 - all naïve to hydroxychloroquine treatment

in Table 2. First-line treatments in the overall patient cohort

The line - All patients received an additional topical treatment with mild corticosteroid or topical calicneurin inhibitors* - 77 (1005)???

4. In the summary, the authors state that the purpose of the study was to evaluate the effectiveness of hydroxychloroquine treatment. Further, the authors state that they studied the effects of hydroxychloroquine, acitretin or other immunomodulatory agents (lines 87-88). Since the authors evaluated the effects of different agents, this should be stated in the summary.

5. Why do the authors refer to hydroxychloroquine as an immunomodulatory agent (line 81)

6. The authors point out that there is a close relation between LLP and the occurrence of Hashimoto thyroiditis. How can the authors explain this relationship in terms of molecular mechanisms?

7. Reference 3 should be properly presented:  Rajan A, Rudnicka L, Szepietowski JC, Lallas A, Rokni GR, Grabbe S, Goldust M. Differentiation of frontal fibrosing alopecia and Lichen planopilaris on trichoscopy: A comprehensive review. J Cosmet Dermatol. 2022 Jun;21(6):2324-2330. doi: 10.1111/jocd.14457.

Author Response

Ad remark 1.In paragraph 2.1, the authors presented a Figure 1. Flow chart illustrating the applied treatment regimens in the LPP patient cohort.  In the text of the article, the authors describe the results of systemic second-line therapy in patients. It would have been better if the authors had supplemented Fig. 1 with a scheme of systemic second-line treatment. This would make the article easier to read.

Ad 1: We thank the reviewer for this important criticism. Thus, we have included an additional flow chart with a scheme of the second line treatment and the response in the patient cohort towards the second line treatment We have included an additional paragraph discussing the limitations of our survey (L229 – 230)

Ad remark 2: In the tables, the research results are presented as a percentage. It is difficult to assess it, as they relate to different groups.

Ad 2: We agree with the point of criticism. Accordingly, we have defined the different groups within the table more clearly, so that it should be easier to interpret the results.

Ad 3: It should be clarified what the authors meant.

Ad 3: We agree with the reviewer that further clarification is need. Thus we have revised the graphs accordingly.

line 22 - (145)???

We apologize for the typo.

line 33 - all naïve to hydroxychloroquine treatment

The sentence has been adapted and is now easier to understand.

in Table 2. First-line treatments in the overall patient cohort The line - All patients received an additional topical treatment with mild corticosteroid or t           opical calicneurin inhibitors* - 77 (1005)???

  We apologize for the typo.

Ad remark 4. In the summary, the authors state that the purpose of the study was to evaluate the effectiveness of hydroxychloroquine treatment. Further, the authors state that they studied the effects of hydroxychloroquine, acitretin or other immunomodulatory agents (lines 87-88). Since the authors evaluated the effects of different agents, this should be stated in the summary.

Ad 4: We agree with the reviewer that that our summary lacked an interpretation of the response towards other systemic agents. Thus, we included a paragraph regarding the repsone to CsA. Due to our limited data regarding the usage of MTX or retinoids we did not include our descriptive data in the summary part ouf our manuscript.

Ad remark 5. Why do the authors refer to hydroxychloroquine as an immunomodulatory agent (line 81).

Ad 5: Hydroxychloroquine and chloroquine are anti-malarial drugs that have been successfully used in the treatment of autoimmune diseases. The main mechanism of action of HCQ is its anti-inflammatory effect. However, by acting on the innate and acquired immune response, HCQ exerts also immunomodulatory effects.4 At the molecular and cellular level, the following 4 mechanisms underly the anti-inflammatory action of HCQ: 1) inhibition of lysosomal activity and autophagy, 2) inhibition of the proinflammatory cytokine signaling pathway, 3) inhibition of nicotinamide adenine dinucleotide phosphate (NADPH) oxidase, and 4) inhibition of the secondary calcium signaling pathway. We integrated passage in our manuscript.

AD remark 6. The authors point out that there is a close relation between LLP and the occurrence of Hashimoto thyroiditis. How can the authors explain this relationship in terms of molecular mechanisms?

AD 6: The molecular mechanisms that link LLP and Hashimoto thyroiditis together are not fully understand yet. However, a possible mode of action is the unmasking of keratinocyte epitopes through thyreoid antibodies. Subsequently, CD8+ cytotoxic T cells recognize the lichen planus antigen associated with MHC class I on lesional keratinocytes and trigger keratinocyte apoptosis All the same, thyroid antibodies linked on the keratinocyte surface may be directly recognized as target antigens by cytotoxic T cells.

AD 7: Reference 3 should be properly presented:  Rajan A, Rudnicka L, Szepietowski JC, Lallas A, Rokni GR, Grabbe S, Goldust M. Differentiation of frontal fibrosing alopecia and Lichen planopilaris on trichoscopy: A comprehensive review. J Cosmet Dermatol. 2022 Jun;21(6):2324-2330. doi: 10.1111/jocd.14457.

AD 7: We have changed the presentation of Reference 3 accordingly.

With kind regards,

Henner Stege

On behalf of all coauthors

Reviewer 2 Report

Comments and Suggestions for Authors

Your efforts are appreciated. 

Regarding the results this section can be improved and summarised.

Tabulate your results as possible and only mention the significant results only 

Any brackets of range must be corrected as it was written in wrong format.

You didn't analyse the causes of treatment failure and on what base the treatment was prescribed and on what base the second line of treatment was given.

Author Response

Reviewer#2

Your efforts are appreciated.

Regarding the results this section can be improved and summarised.

Tabulate your results as possible and only mention the significant results only

Any brackets of range must be corrected as it was written in wrong format.

You didn't analyse the causes of treatment failure and on what base the treatment was prescribed and on what base the second line of treatment was given.

AD Reviewer #2: We thank the reviewer for his essential and valuable criticism and feedback. Accordingly, we have summarized and tabulated our results, focusing mainly on our significant results. However, we left certain non-significant results in our manuscript since these findings are important for the further treatment of LLP, e.g., our retrospective analysis did not show a correlation between the response towards the systemic treatment and the time from diagnosis to treatment.

Furthermore, we have meticulously rearranged all range brackets, ensuring the accuracy and clarity of our data presentation.

Due to the lack of patient material, we were not able to investigate the causes of treatment failure further. We further clarified on what basis the systemic agent (first or second line) was prescribed. Overall, we initiated systemic treatment only in patients with no response to topical treatment. Due to the slow progression of the disease, treatment was then initiated when patients felt a worsening of clinical symptoms.  

Reviewer 3 Report

Comments and Suggestions for Authors

A clinical diagnosis is not the same as a biopsy report. The latter makes it possible to accurately establish the diagnosis and the degree of activity. It would be preferable if all the patients studied had been biopsied, so that the sample is uniform.

Authors should specify dosage and schedule of administration for topical treatment and dosage and schedule of administration for systemic treatment

Author Response

Reviewer 3

AD remark 1: A clinical diagnosis is not the same as a biopsy report. The latter makes it possible to accurately establish the diagnosis and the degree of activity. It would be preferable if all the patients studied had been biopsied, so that the sample is uniform.

AD 1: We agree with the reviewer that a clinical diagnosis is not equal to a biopsy report in most cases. However, the diagnosis of Lichen planopilaris or Frontal fibrosing alopecia always involves an interplay between the clinical manifestation (patchy scarring alopecia with itching and burning, follicular hyperkeratosis, and perifollicular erythema) alongside the histological findings of an interface dermatitis. The histological presentation of interface dermatitis is not pathognomonic for LLP/FFA but is also common in lupus erythematosus, dermatomyositis, or graft vs. host disease. Thus, we believe inclusion into our patient cohort is possible when the patient presents the typical clinical symptoms without a corresponding biopsy report.

AD remark 2:Authors should specify dosage and schedule of administration for topical treatment and dosage and schedule of administration for systemic treatment.

AD 2: We thank the reviewer for this vital hint, and accordingly, we have added the dosage of the systemic agents as well as the application of the topical treatment.